# Is Two Better Than One? The Impact of Doubling Training Volume in Severe COPD: A Randomized Controlled Study

**DOI:** 10.3390/jcm8071052

**Published:** 2019-07-18

**Authors:** Mara Paneroni, Ioannis Vogiatzis, Stefano Belli, Gloria Savio, Dina Visca, Elisabetta Zampogna, Maria Aliani, Vito De Carolis, Mauro Maniscalco, Carla Simonelli, Michele Vitacca

**Affiliations:** 1Istituti Clinici Scientifici Maugeri IRCCS, Respiratory Rehabilitation of the Institute of Lumezzane, 25065 Lumezzane (BS), Italy; 2Faculty of Health and Life Sciences, Department of Sport, Exercise and Rehabilitation, Northumbria University, Newcastle NE1 8ST, UK; 3Istituti Clinici Scientifici Maugeri IRCCS, Respiratory Rehabilitation of the Institute of Veruno, 28010 Veruno (NO), Italy; 4Istituti Clinici Scientifici Maugeri IRCCS, Respiratory Rehabilitation of the Institute of Tradate, 21049 Tradate (VA), Italy; 5Istituti Clinici Scientifici Maugeri IRCCS, Respiratory Rehabilitation of the Institute of Cassano delle Murge, 70020 Cassano delle Murge (BA), Italy; 6Istituti Clinici Scientifici Maugeri IRCCS, Respiratory Rehabilitation of the Institute of Telese, 82037 Telese Terme (BN), Italy

**Keywords:** COPD, exercise training, pulmonary rehabilitation

## Abstract

Patients with severe chronic obstructive pulmonary disease (COPD) are unable to exercise at high intensities for sufficiently long periods of time to obtain true physiological training effects. It therefore appears sensible to increase training duration at sub-maximal exercise intensities to optimize the benefit of exercise training. We compared the effects on exercise tolerance of two endurance cycloergometer submaximal exercise protocols with different cumulative training loads (one (G1) versus two (G2) daily 40 min training sessions) both implemented over 20 consecutive days in 149 patients with COPD (forced expiratory volume at first second (FEV_1_): 39% predicted) admitted to an inpatient pulmonary rehabilitation program. Patients in G2 exhibited greater improvement (*p* = 0.011) in submaximal endurance time (from 258 (197) to 741 (662) sec) compared to G1 (from 303 (237) to 530 (555) sec). Clinically meaningful improvements in health-related quality of life, 6MWT, and chronic dyspnea were not different between groups. Doubling the volume of endurance training is feasible and can lead to an additional benefit on exercise tolerance. Future studies may investigate the applicability and benefits of this training strategy in the outpatient or community-based pulmonary rehabilitation settings to amplify the benefits of exercise interventions.

## 1. Introduction 

Exercise training is the cornerstone of pulmonary rehabilitation, with endurance training representing the most commonly implemented modality in this setting for patients with chronic obstructive pulmonary disease (COPD). The benefits of endurance training in this population are well documented and include reductions in hospitalization and exacerbation rates, sensations of breathlessness during daily activities, and improvements in exercise tolerance, functional capacity, and health-related quality of life [1].

In outpatients with COPD, recommended training programs consist of a minimum of 16–20 sessions, each lasting for 30–90 min, that are performed 3–5 times per week, at an intensity equivalent to 60–80% of peak exercise capacity (Wpeak) [2]. In inpatients with COPD, training frequency may increase to 5 times per week with similar loading characteristics as for outpatients with COPD [3].

In healthy young individuals there is strong evidence that the higher the endurance training volume, the greater the magnitude of physiological adaptations [4]. In older healthy people, there is, however, a less clear dose–response effect for endurance exercise training volume [5,6].

In patients with COPD, there are studies comparing the physiological benefits of different endurance training intensities [7,8,9], within or across different exercise training modalities (continuous vs. interval training) [10,11,12]. The optimal training intensity that is tolerable by the majority of patients and that yields the greatest physiological benefits is still unclear [13].

Exercise training volume refers to the product of intensity applied over a given duration during a bout of exercise and is used to calculate the training dose of exercise. Training volume may be increased by increasing the exercise duration, the exercise intensity, or both [14]. However, severe COPD patients are unable to exercise at high intensities for sufficiently long periods of time to obtain true physiological training effects due to central cardiopulmonary and peripheral muscular limitations [15,16]. It therefore seems sensible to increase training duration at a given sub-maximal intensity in these patients to optimize the benefit of exercise training. This approach is feasible and may elicit greater physiological benefits compared to the standard training programs outlined in the joint American Thoracic Society/European Respiratory Society (ATS/ERS) statement on pulmonary rehabilitation [2].

Accordingly, the aim of this pragmatic randomized controlled trial was to compare the effects on exercise tolerance of two endurance cycloergometer exercise protocols with different cumulative training loads (one versus two daily 40 min training sessions) implemented over 20 consecutive days in severe COPD patients admitted to an inpatient pulmonary rehabilitation program. We also assessed the effects of the two programmes on respiratory and peripheral muscle strength, symptoms, health-related quality of life, and the acceptability and adherence to the different training programmes (patients’ satisfaction, dropouts, and side effects) and explored potential associations between the baseline patient characteristics and the magnitude of improvement in exercise tolerance following exercise training. We hypothesized that two daily training sessions would be superior to a single daily session in terms of improving exercise tolerance. 

## 2. Experimental Section

### 2.1. Study Population

Eligible participants were consecutive stable inpatients with severe COPD, diagnosed according to the Global Initiative for Lung Diseases criteria [17] and referred by their physicians to pulmonary rehabilitation. Inclusion criteria were as follows: forced expiratory volume at first second/forced vital capacity (FEV_1_/FVC) <70% and FEV_1_ <50% of predicted values and clinical stability (no changes in medication within the previous 10 days). Exclusion criteria were as follows: (1) participation to a program of pulmonary rehabilitation including exercise training within the previous six months, (2) the necessity of modification of drug therapy within the previous 15 days, (3) a recent acute myocardial infarction within the previous three months, (4) a presence of chronic heart failure, (5) severe musculoskeletal conditions limiting the ability to perform regular exercise, (6) previously reported psychiatric conditions, and (7) cognitive impairments (mini-mental state examination <22) [18]. When exacerbations requiring a strong modification of drug therapy (a need for antibiotic and/or oral steroid) occurred during the course of the study, patients discontinued their participation to the study. Data from patients exhibiting a low compliance with the programme (i.e., missing more than five consecutive days of training sessions since the onset of the program) were excluded from the analysis. All institutes provided ethics approval for the trial (protocol number 1066 Ethics Committee, session of 11 May 2015), and all participating patients gave their written informed consent. The trial was registered on the clinicaltrials.gov website (NCT02522637).

### 2.2. Study Design

This was a prospective, multicenter, two parallel groups (one versus twice daily training), randomized controlled trial. The study was carried out at the Pulmonary Rehabilitation Department of the Istituti Clinici Scientifici Maugeri IRCCS, Pavia, Italy. The centers that were involved in the study were as follows: Lumezzane (BS), Pavia, Cassano delle Murge (BA), Telese Terme (BN), Tradate (VA), and Veruno (NO). All patients included in the study undertook a comprehensive pulmonary rehabilitation program that was delivered by experienced multidisciplinary teams consisting of physicians, physiotherapists, nurses, physiologists, and other health care professionals. The physiotherapist who conducted the initial and final assessments was blinded on the patient’s allocation. Patient blinding was not possible due to the nature of the study, and the physiotherapist who supervised the training was not blinded. Following baseline assessment, patients were randomized according to a computerized randomization list (www.randomization.com) with a block design and an allocation ratio of 1:1. A data manager, who kept and consulted the randomization list, communicated patients’ allocation to the physiotherapists responsible of the study in each center that was blinded to the randomization. Randomization was stratified by FEV_1_ (<50% predicted or >50% predicted).

### 2.3. Assessments

At enrolment, baseline anthropometrical and clinical data were collected. Comorbidities were assessed by the Cumulative Illness Rating Scale (CIRS) scale [19]. At baseline and at the end of the training program, all patients underwent the following assessments:The Six-Minute Walking Test (6MWT), performed according to ATS/ERS [20]. Supplemental oxygen, when needed (SpO_2_ < 90%), was delivered at the flow prescribed by the physician and was kept at the same level during both assessment time points. Predictive values were calculated according to Chetta et al. [21]. We evaluated meters walked and percentage of patients that improved above the minimal clinical important difference (MCID) of 30 m [20].The Constant Work Rate Exercise Test (CWR) (primary outcome) [22], cycling with a load set at 70% of the individual maximum predicted load, calculated according to the equation of Luxton et al. [23]. Patients pedaled at 50–60 revolutions per minute at constant load until their limit of tolerance (reaching Borg dyspnea or fatigue ≥8, or a ≥90% maximum theoretical heart rate). Endurance time to the limit of tolerance (in seconds) was measured, and pre-to-post variation (before and after the training program) was calculated. We also defined the percentage of patients with an improvement above the minimal clinical important difference (MCID) of 105 s [22].Maximal inspiratory pressure (MIP) and maximal expiratory pressure (MEP), collected at the mouth in the seated position were performed according to the ATS/ERS indications [24]. Predicted values were calculated according to Black and Hyatt [25].Maximal voluntary contraction (MVC) of the quadriceps muscles was carried out using a hand-held dynamometer (Chatillon DMG-200, Ametek, Largo, FL, USA). Body positions for the tests were standardized, and predictive values were calculated using the method proposed by Andrews et al. [26].Medical Research Council (MRC) dyspnea, on a 0–5 scale, with 0 corresponding to no dyspnea and 5 to the worse level of dyspnea [27].Health-related quality of life by the COPD Assessment Test (CAT), on a 0–40 scale, with 40 corresponding to the greatest impact of COPD on quality of life [28], by the Maugeri Respiratory Failure modified scale (MRF-26) [29].

In addition, at the end of the program, patients’ perceived satisfaction with regard to “quality of training perception” was investigated using a 0–4 Likert scale [30], with the following answers: 0 = very bad; 1 = bad; 2 = sufficient; 3 = good; 4 = very good.

Perception of “quantity of training” was tested by the following question: “How do you define the amount of the training performed?” with three possible answers: 0 = not sufficient; 1 = good; 2 = too much. Side effects during training and drop-out rates were monitored. 

### 2.4. Interventions

Patients performed a comprehensive inpatient pulmonary rehabilitation program lasting for 20 consecutive days including exercise training, psychological support, medical assessment, educational components of COPD management, and chest physiotherapy when necessary. The exercise program included endurance training and active mobilization exercises as follows.

#### 2.4.1. Endurance Training

Patients enrolled were randomized into two groups and undertook one of the following exercise interventions:Group 1 (G1) underwent a single daily 40-min session carried out in the morning or in the afternoon according to individual preferences for 20 consecutive days (20 sessions in total).Group 2 (G2) underwent two daily 40-min sessions for 20 consecutive days (40 sessions in total) comprising one session in the morning and one session in the afternoon; sessions were separated by at least three hours of rest in between.

Each session of training consisted of 40 min of cycling at moderate-high intensity (30 min of cycling at a constant load, a 5-min warm-up, and a 5-min cool-down without load). Starting load intensity was set at 50% of the maximum predicted load in watts according to the equation of Luxton et al. [23] based on gender, age, and distance walked during the 6MWT aiming for dyspnea and/or leg discomfort scores equivalent to 4–5 on the Borg scale. Management of intensity progression in subsequent sessions was carried out based on symptoms of dyspnea and fatigue on the Borg scale [31] previously described by Maltais et al. [16]. Briefly, we increased the load by 10 watts per session when in the preceding session patients defined their dyspnea and/or leg discomfort as less than 4–5 on the Borg scale; the workload was decreased if dyspnea or leg discomfort scores were greater than 4–5 or the heart rate exceeded 90% of the predicted maximal heart rate (220 minus age).

Oxygen pulsoximetry (SpO2), blood pressure, perceived dyspnea and leg discomfort on the Borg scale were measured at the beginning and at the end of each training session; heart rate was continuously monitored by telemetry.

#### 2.4.2. Active Mobilization Exercises

Following each 40 min cycling session, all patients underwent a 20-min session including the following exercises supervised by a physiotherapist: cool-down and group active mobilization, mild-intensity strengthening, and free-body exercises using lightweight, sticks, and balls. 

Medication was optimized before starting the training program and any changes were recorded during the study period. During exercise, the use of oxygen therapy was allowed in case of patients showing SpO2 values <90%, and it was delivered at a flow sufficient to maintain an SpO2 value >90%.

### 2.5. Statistics

To detect the sample size, the hypothesis was that the primary endpoint (the endurance time during constant work rate exercise testing: CWR) improved by a minimum of 120 s more in G2 as compared to G1 [9]. Considering a 1:1 randomization ratio, a study power of 90% where *p* < 0.05 and hypothesizing a standard deviation of a primary endpoint of 200 s, a minimum total sample size of 118 patients was calculated. Because of the severity of the population, we took into account a 25% dropout rate and increased our total sample size by 30, estimating a total sample size of 148 patients.

Statistical analysis was conducted by statistical software STATA 13 (StataCorp, LP, College Station, TX, USA).

Descriptive statistics of all measured variables were performed indicating mean and standard deviation for continuous variables and frequency distributions for categorical and ordinal variables. Analytical statistics tested the pre-to-post difference of response in the two groups by *t*-tests. We compared the different rate of patients who improved the 6MWT and the CWR by Pearson Chi^2^ test and evaluated the risk of improvement by odd ratio analysis using the Improvers/Not Improvers groups (see above, in the Assessments section) as variables dependent and baseline characteristics (FEV_1_%, FVC%, Residual Volume (RV)%, FEV_1_/FVC%, age, sex, body mass index (BMI), chronic respiratory failure (CRF), partial pressure of arterial oxygen/inspiratory fraction of oxygen (PaO2/FiO2), partial pressure of arterial carbon dioxide (PaCO2), 6MWT, CWR) and group allocation as independent variables. Trends of workload, heart rate, and symptoms during training were analyzed by mixed effect analysis of variance (ANOVA) with time and group as factors. All statistical tests were considered significant when *P* < 0.05. 

## 3. Results

### 3.1. Study Population

From June 2015 to May 2018, 950 COPD patients were evaluated to our pulmonary rehabilitation (PR) program. Seven hundred ninety-one did not fulfill the inclusion criteria and were excluded, 10 refused to participate, and 149 were enrolled. Seventy-eight (78) were allocated to Group 1 and 71 to Group 2. During the time course of the study, a total of 31 (20.8%) patients discontinued the programme, while 118 patients completed the study. Reasons for discontinuing the programme were as follows: refused to continue (3), transferred to another hospital (6), hemoptysis (1), respiratory exacerbation (5), severe arrhythmias (7), severe muscle soreness (5), no compliance (3), and heart attack (1). The trial profile of the study (CONSORT diagram) is illustrated in Figure 1.

Table 1 describes the anthropometric and clinical characteristics of the patients studied.

Patients enrolled exhibited severe obstruction, with a high degree of lung hyperinflation. They had a moderate dyspnea during daily living and more than half of them were on long-term oxygen therapy. Exercise tolerance and quality of life were profoundly compromised (Table 1).

### 3.2. Intervention Results

Figure 2 presents a box plot of changes in the primary outcome (Time to Limitation (Tlim) for CWR), while Table 2 describes pre-to-post changes in secondary outcomes of the study and the differences between groups (mean difference (95% Confidence Interval) = 252 (54–451)).

The primary outcome (cycloergometer endurance time) improved more in G2 (by 483 sec) (from 258 (197) to 741 (662) sec) than in G1 (by 227 sec) (from 303 (237) to 530 (555) sec); (*p* = 0.0113), whereas all the other outcome measures showed no significant differences in the magnitude of improvement between the two groups. Interestingly, the percentage of improvers in cycloergometer endurance time in G2 was greater than that in G1 (Figure 3). Quadriceps muscle strength was not improved in neither of the groups, and MIP did not improve in G2.

The allocation to G2 doubled the prospect of improvement in exercise tolerance (seconds) during CWR (OR 2.43; 95% IC 1.073–5.513, *p* = 0.031) and in walking distance (meters) during the 6MWT (OR 2.50; 95% IC 1.165-5.362, *p* = 0.019) compared with the allocation to G1. No other baseline variables affected the improvement in cycling endurance time, while improvement in the 6MWT was related to CRF presence (OR 0.4193, IC 95% 0.1949–0.9038, *p* = 0.027) and the 6MWT baseline (OR 0.9930, IC 95% 0.9885-0.9975, *p* = 0.002), the hypoxemic and the more disabled patients being more likely to improve. 

Figure 3 shows that, when compared to G1, G2 included more patients who improved above MCID during both the cycling test (*p* = 0.043) and the 6MWT (*p* = 0.045). 

Patients’ satisfaction was similar between groups with regard to quality perception of the rehabilitation program (G1: 2 = 13.79%; 3 = 55.17%, 4 = 31.03% versus G2: 2 = 10.91%; 3 = 45.45%, 4 = 43.62%. *p* = 0.382) and the amount of training load (G1: 0 = 6.56%; 1 = 85.25%, 2 = 8.2% versus G2: 0 = 3.51%, 1 = 84.21%, 2 = 12.28% *p* = 0.0599). 

### 3.3. Training Program Progression

The initial training intensity on the cycloergometer was 26.33 (10.55) watts in G1 and 26.41 (12.37) watts in G2 (*p* = 0.097), whereas the magnitude of improvement in cycling total training workload from the initial training session was greater in G2 (increase by 114 (121)%) compared to G1 (increase by 83 (82)%); (*p* = 0.0179)). The day-by-day trend of modification of workload, heart rate, and lower recorded symptom scores for dyspnea and leg discomfort is presented in Figure 4. 

## 4. Discussion

The major finding of the study is that in patients with severe COPD doubling the volume of endurance training (twice daily as opposed to once daily) has led to an additional benefit on cycling exercise tolerance following the completion of an inpatient pulmonary rehabilitation programme. Other clinical and functional outcome measures significantly and meaningfully improved, with the fraction of patients exhibiting clinically meaningful improvements in both exercise capacity measures (cycloergometer endurance time and the 6MWT), which were significantly greater for those patients trained twice daily. Furthermore, patients assigned to the twice-daily exercise training group well adhered to the increased physical training demands of the programme and afforded a 100% increase in overall training workload. Therefore, our findings suggest that in inpatients with severe COPD, increasing the daily volume of moderate-intensity exercise training is well tolerated and yields additional benefits on exercise capacity. This finding is of clinical significance when considering pulmonary rehabilitation in the inpatient setting. 

Traditional approaches and guidelines of exercise prescription in patients with COPD recommend a target training load above 60% of peak exercise capacity, with progressive increases over time and a frequency ranging from 3 to 5 sessions per week, with a minimum of 20 sessions [13,32]. However, as it has been known for quite some time, few patients with severe COPD can tolerate training intensities at 60–80% peak capacity for 30 consecutive minutes; typically these severe COPD patients are capable of progressively sustaining exercise intensities with a peak capacity between 50 and 60% [16]. Our approach to provide patients with such tolerable exercise intensities for prolonged periods of time on a daily basis over a course of 20 days has proven both feasible and effective even for those patients who were required to exercise at moderate intensities twice daily. Furthermore, our approach is in line with approaches of training prescription which have been successfully applied in healthy sedentary and trained individuals [33].

Our study is the first of its kind in testing the possibility to regularly exercise COPD patients with a high training volume (ten 30-min sessions a week), while improving important patient outcomes. In line with our results, recently Morris et al. suggested that the impact of training volume can influence the physiological improvement more that the intensity applied, suggesting a need for further investigation into the “dose” of endurance training [13]. Our findings thus provide a novel element regarding the tailoring of training prescription, beyond the current training recommendations [2].

The present study shows that both groups exhibited a clinically meaningful improvement in constant-load cycling endurance time following the completion of the training programmes. However, patients who exercised twice daily demonstrated significantly greater improvement in cycling endurance time compared to those who trained once daily. The four-fold difference in the magnitude of improvement in endurance time between the two groups exceeds the clinically meaningful difference (i.e., a 33% improvement in endurance time) described for pharmacological and non-pharmacological interventions in patients with COPD [22]. In both groups, the improvement in the 6MWT exceeded the clinically meaningful margin of 30 m, but the fraction of patients exhibiting clinically meaningful improvements in the 6MWT was significantly greater for those patients trained twice daily. This finding reinforces the notion that the effects of exercise training are strictly related to the amount of training workload applied [34,35,36,37].

Furthermore, evidence of true physiological training adaptations for both groups is provided by the greater increase in the sustained cycling work rate throughout the programme that was achieved by the cardiovascular responses and sensations of breathlessness and leg discomfort comparable to those who trained once daily (Figure 2). This suggests that during the rehabilitation programme exercise fitness levels improved in both groups as work rates increased, while cardiac response and symptoms remained relatively unchanged and similar between groups [12].

Interestingly, the clinically meaningful improvement in the 6MWT was highly comparable between groups. This finding most likely reflects the inability of severe COPD patients to increase walking speed, which is known to be compromised in this population [38]. In fact, the post-training 6MWT of approximately 410 m recorded in both groups reflects an average walking speed of approximately 1.1 m/sec, which represents the upper boundaries of walking speed in severe COPD patients [38,39]. In addition, it should be emphasized that training adaptations are specific to training modality employed [38]; hence, one would expect cycling training adaptations to be better reflected during cycling testing than walking testing. This may also explain the finding that the addition of a second training session on a daily basis, as compared to only one daily training session, did not have significant additive effects on respiratory or peripheral muscle strength.

Similarly, both training modalities induced comparable improvements in other clinical outcomes, namely MRC, CAT, and MRF scores. A lack of significant differences in the magnitude of improvement in these outcomes is most likely attributed to the multidisciplinary component of the comprehensive pulmonary rehabilitation programme; this one was designed to improve, in addition to exercise tolerance, symptoms and quality of life. Hence, doubling the number of sessions during inpatient rehabilitation conveys superior effects on exercise tolerance along with clinically meaningful improvements in symptoms and quality of life.

### 4.1. Clinical Implications

When compared to international recommendations for pulmonary rehabilitation [2], our findings suggest that a higher volume of training could be applied in the setting of a short-term inpatient pulmonary rehabilitation program to reach higher levels of exercise tolerance and that it meaningfully improved quality of life and reduced sensations of breathlessness. Hence, training dosage is important during pulmonary rehabilitation and can have a substantial effect on the magnitude of exercise-induced physiological adaptations. While resources available to other pulmonary rehabilitation centers globally may limit the applicability of our approach to offer two training sessions per day, our findings are highly pertinent to these National Health Systems that provide a short period of inpatient rehabilitation: in this case, patients may be encouraged to attend two training sessions per day in an attempt to boost their fitness levels. 

### 4.2. Limitations 

We acknowledge that our study has a number of limitations. We did not perform a Cardiopulmonary Exercise Test (CPET) at the outset of the study to precisely set the training intensity during exercise training. Instead, we defined the initial training load via estimation from the 6MWT [23]. This procedure could have underestimated or overestimated the initial training load with respect to the individual patient exercise capacity [40]. Nevertheless, this procedure reflects the “real life” assessment commonly used in the pulmonary rehabilitation settings involved in the present study where access to time and recourse demanding CPET is not available. This was a pragmatic trial reflecting the resources of the average pulmonary rehabilitation center in Italy where sophisticated and costly CPET measurements are not available. Furthermore, the lack of CPET measurements has not allowed the undertaking of physiological measurements to better appreciate the physiological benefits of the two training interventions. Lastly, the very tight patient exclusion criteria implemented in the present investigation produced a large sample of excluded patients, thereby limiting the external validity of the rehabilitation trial. However, in the real-life setting, we are confident that some of the implemented exclusion criteria could be avoided (i.e. pulmonary rehabilitation in the previous six months or stability in chronic heart failure) without an adverse impact on safety or efficacy.

## 5. Conclusions

The major finding of this study is that, in severe COPD patients, doubling the volume of endurance training (twice daily as opposed to once daily) is feasible and can lead to an additional benefit on cycling exercise tolerance following the completion of an inpatient pulmonary rehabilitation programme. Future studies may investigate the applicability and benefits of this training strategy in the outpatient or community-based pulmonary rehabilitation settings to amplify the benefits of exercise interventions. 

## Figures and Tables

**Figure 1 jcm-08-01052-f001:**
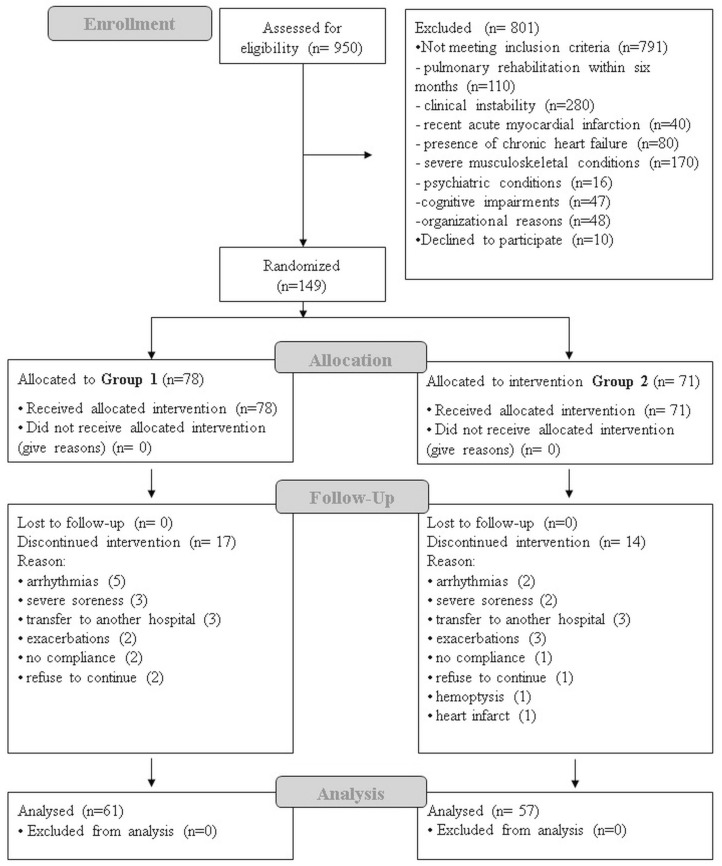
CONSORT Trial flow of the study.

**Figure 2 jcm-08-01052-f002:**
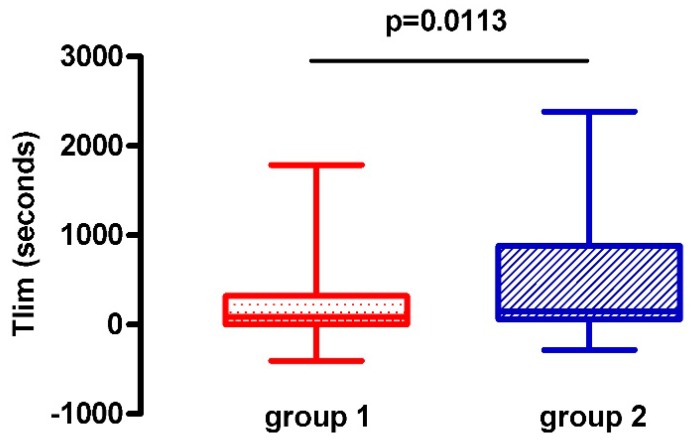
Pre-to-post changes in endurance time of CWR (primary outcome).

**Figure 3 jcm-08-01052-f003:**
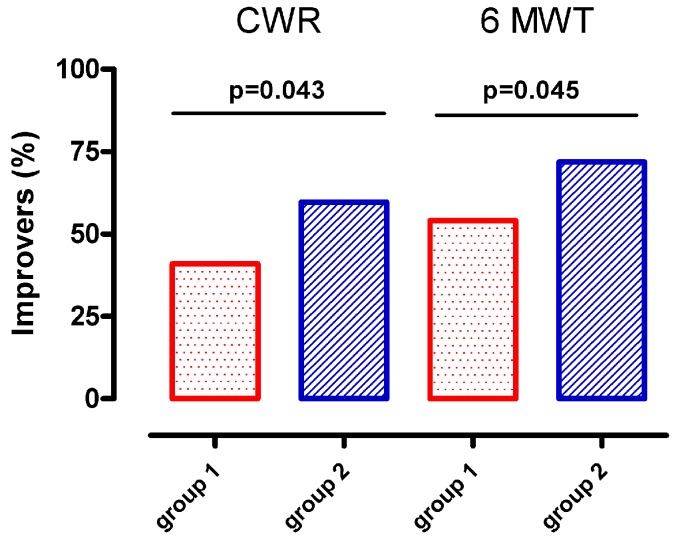
Fraction of improvers above MCID CWR (105 s) [22] and in 6MWT (30 m) [20] tests.

**Figure 4 jcm-08-01052-f004:**
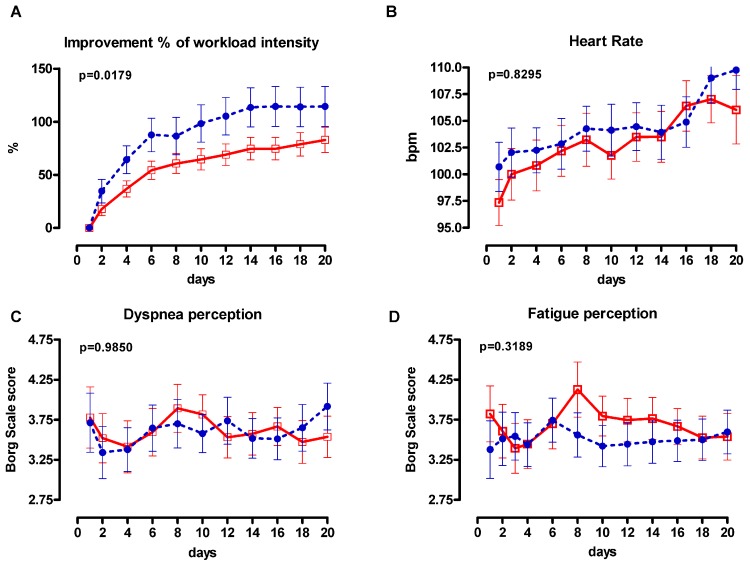
**A**: Day by day workload (expressed percentage improvement from the initial training session), **B**: heart rate and **C**/**D** symptoms during training. Closed circles = twice daily (G2); open squares = once daily (G1). The values for G2 are the average of maximum values over the two daily training sessions. Error bars indicate standard error (SE) of the mean. P refers to the ANOVA time group factor.

**Table 1 jcm-08-01052-t001:** Anthropometric and clinical characteristics of included patients.

MEASURES	Group 1 (n = 78)	Group 2 (n = 71)	P
**Sex**, male/female	60/18	53/18	0.8200
Age, years	69 (9)	69 (8)	0.9818
BMI, Kg/m^2^	25.53 (5.46)	25.61 (4.09)	0.9118
FEV1, %predicted	39.54 (11.93)	39.56 (11.36)	0.9303
FVC, %	75.03 (19.73)	75.32 (17.26)	0.9428
FEV1/FVC	43.94 (11.41)	42.75 (10.20)	0.4930
RV, %	180 (56)	180 (53)	0.7619
MIP, cmH2O	66.59 (22.28)	69.43 (25.22)	0.2517
MEP, cmH2O	90.33 (35.77)	85.73 (32.88)	0.4822
LTOT, %	60	59	0.7940
CIRS, 1st item, score	2.27 (2.92)	2.09 (3.16)	0.6939
CIRS, 2nd item, score	2.51 (1.42)	2.49 (1.61)	0.8131
PaO2/FiO2	310 (53)	308 (45)	0.5410
PaCO2, mmHg	40.47 (6.31)	40.54 (5.63)	0.7083
pH	7.42 (0.03)	7.43 (0.03)	0.0951
6MWT, meters	376 (92)	358 (88)	0.4202
CWR, s	288 (226)	246 (186)	0.2592
MRC, score	3.31 (4.13)	2.57 (1.07)	0.1630
MVC, quadriceps, Kg	24 (7)	24 (9)	0.9415
MRF26, score	11.53 (6.811)	9.98 (5.97)	0.2058
CAT, score	21 (7.25)	20.33 (6.76)	0.7762

Legend: % = percentage, FEV1 = forced expiratory volume 1, FCV = forced vital capacity, RV = residual volume, MIP = maximal inspiratory pressure, MEP = maximal inspiratory pressure, LTOT = long-term oxygen therapy, CIRS = Cumulative Illness Rating Scale, PaO2 = arterial pressure of oxygen, FiO2 = inspiratory fraction of oxygen, PaCO2 = arterial pressure of carbon dioxide, 6MWT = Six-Minute Walking Test; CWR = Constant Work Rate Exercise Test, MRC = Medical Research Council Dyspnea score, MRF26 = Maugeri Respiratory Failure-26 Scale, CAT = COPD Assessment Test.

**Table 2 jcm-08-01052-t002:** Pre-to-post changes in secondary outcomes.

	Group 1 (n = 61)			Group 2 (n = 57)				
	Baseline	At the end	P Pre-to-post	Baseline	At the end	P Pre-to-post	Differences between Groups (G2-G1) Mean (IC 95%)	P between Groups
**6MWT**, meters	373 (89)	416 (89)	0.001	360 (93)	410 (99)	0.001	7.953 (−14.44, 30.35)	0.4832
**MIP**, cmH2O	67 (23)	74 (23)	0.0471	72 (26)	76 (20)	0.2538	−3.05 (−11.8, 5.7)	0.4900
**MEP**, cmH2O	92 (39)	104 (47)	0.040	86 (33)	99 (34)	0.0002	0.939 (−12.36, 14.23)	0.8887
**Quadriceps**, Kg	24.4 (7.2)	25.1 (7.5)	0.5164	25 (9.2)	27 (10.9)	0.2690	1.29(−2.90, 5.48)	0.5400
**MRC**, score	3.6 (4.9)	1.7 (1.2)	0.007	2.7 (1.1)	1.5 (1)	0.0001	0.763(−0.07, 2.20)	0.2960
**CAT**, score	21.5 (7.6)	14.4 (7.8)	0.001	21.1 (7.1)	11.9 (7)	0.0001	−2.03(−5.08,1.02)	0.1900
**MRF**, score	11.53 (6.8)	8.22 (6.7)	0.001	9.98 (5.97)	7.28 (5.81)	0.0001	0.6122 (−1.20, 2.43)	0.5049

Legend: 6MWT = Six-Minute Walking Test; MIP = maximal inspiratory pressure; MEP = maximal inspiratory pressure; MRC = Medical Research Council Dyspnea score; CAT = COPD Assessment Test; MRF26 = Maugeri Respiratory Failure-26 Scale; IC 95% = 95% confidence interval (lower value, upper value).

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
