# Peer review of "Is Two Better Than One? The Impact of Doubling Training Volume in Severe COPD: A Randomized Controlled Study"

_jcm, 2019, doi:10.3390/jcm8071052_

Reviewer 1 Report

I kindly suggest only to try to add new references at discussion section.

Reviewer 2 Report

The study addresses an important topic and has a clear rationale for the research question. It finds an additional benefit of doubling the number of training sessions on endurance time on an endurance cycling test but does not discover benefits on secondary outcomes. The authors provide a sound rationale for the study and have overall a good discussion about the study results and its implication. There are some issues that I believe need to be addressed.

 ...................

Abbreviations: All abbreviations should be written in full the first time mentioned in the text, for example, lines 37, 109, etc. For abbreviations mentioned << span=""> 3 times, I would recommend not to use an abbreviation and write it in full.

 Language, punctuations, etc.: I would recommend that the manuscript is proofread by an individual with English as their first language / or the paper is sent to language editing.

 Line 19: Remove the A in front of Patients.

 Line 62/92: The authors state that the trial was a pragmatic randomized controlled trial. I would therefore recommend that authors fill out and attach the checklists for these guidelines to ensure

that all necessary items within these checklists are addressed within the submitted article.

 Line 116: Isn't constant work rate (CWR) test the more commonly used description of a constant load exercise test?

 Line 116-123. Issues with the references? E.g., the authors state that they performed the CLET test according to ATS indications. But they refer (reference 23) to ATS statement on Respiratory Muscle Testing? The same for the MCID of 46 seconds, they refer to a study by Jones et al. on the CAT test?

 Line 123. Is the MCID for the CLET really 46 seconds? I believe it is (∼100 s in the 75% CLET test, and ∼70 s in the 85% CLET test)

 Line 169: Resistance exercises. Please describe more in detail about the resistance exercises in a similar way as the endurance protocol. For example; load, repetitions, type of exercise, progression etc. It is of particular interest due to the lack of effect on quadriceps strength in both groups. I would encourage the authors to follow either the Template for Intervention Description and Replication and the Consensus on Exercise Reporting template for reporting intervention studies. It will assist in providing the necessary information.

 Line 184 to 194. Statistical analysis. I would strongly recommend the authors to perform an intention to treat analysis instead or in addition to this per-protocol analysis. Could the authors further explain why an ITT wasn’t used?

 Line 197. Since exclusion rates were > 80%, I would like the authors to discuss what impact this has on the external validity of their findings.

 Table 2 and figure 2: I would suggest that authors, in addition to p-values or instead of p-values, also add information on mean difference and 95% confidence intervals.

 Figure 2: Is the scale correct? Did you have patients in the intervention group that did > 40 minutes longer on the CLET test after training? Was no upper time limit used?

 Figure 3. If I understand the figure correctly, the authors used 54 meters as MCID, but I do believe that 30 meters are considered the MCID for the 6 MWT? Please redo the analysis based on a 30-meter MCID.

 Figure 3. With regard to the MCID for the CLET test, is it 46 or 105 seconds?  

 Line 248 to 255 and Figure 4. The rationale for the study was that doing 40 vs. 20 sessions would result in larger adaptations due to a larger total workload. However, this is not put forward in the section on training program progression. In Figure 4, the values for G2 are the average of maximum values over the two daily training sessions illustrating that the work per session was not different between G1 and G2. However, I would recommend illustrating that the total training workload was higher in G2. Also, I would recommend demonstrating the magnitude of the difference in total workload between G1 and G2.

 Figure 4: The rate of progression was not different between the two groups; the mean workloads were similar. Why is that? Since G2 performed twice as many sessions as G1, shouldn’t it be logical that they would have been able to reach higher intensities? Especially since Figure 4A does not demonstrates a continued progression over the intervention period (no plateau).

 Figure 4. On line 143 do the authors describe that the training lasted for 20 consecutive days but in Figure 4 are only 15 days reported?

 Line 318-319. Use the reference number of Spruit 2013 instead of writing “Spruit et al. 2013.”

 The discussion is mainly focused on the positive effects of the study on the primary outcome. However, even though it is logical to focus on the primary outcome, the lack of between-group difference in secondary outcomes should be further addressed. Especially as effects on 6MWT, CAT MRF, etc. (even though being secondary outcomes in this study) is perhaps even more important for the prognosis and quality of life of the individual patient? The authors have a good rationale related to the principle of specificity about the lack of effect on the 6MWT. However, if doubling the number of sessions do not lead to greater effects on outcomes other than endurance time on a CLET test, should we still double the number of sessions in our patients?

 Author Response
